# Serum Untargeted Metabolism Reveals the Mechanism of *L. plantarum* ZDY2013 in Alleviating Kidney Injury Induced by High-Salt Diet

**DOI:** 10.3390/nu13113920

**Published:** 2021-11-01

**Authors:** Cuixiang Wan, Shufang Chen, Kui Zhao, Zhongyue Ren, Lingling Peng, Huiling Xia, Hua Wei, Bo Yu

**Affiliations:** 1State Key Laboratory of Food Science and Technology, Nanchang University, Nanchang 330047, China; cuixiangwan@ncu.edu.cn (C.W.); shufangchen2021@163.com (S.C.); zk17770843514@163.com (K.Z.); zhongyueR666@163.com (Z.R.); penglingling6@163.com (L.P.); weihua@ncu.edu.cn (H.W.); 2Sino-German Joint Research Institute, Nanchang University, Nanchang 330047, China; hlxia_8798@126.com

**Keywords:** high-salt diets, intestinal flora, kidney injury, *Lactobacillus*, untargeted metabolism

## Abstract

A high-salt diet (HSD) is one of the key risk factors for hypertension and kidney injury. In this study, a HSD C57BL/6J mice model was established with 4% NaCl, and then different concentrations of *Lactobacillus plantarum* ZDY2013 were intragastrically administered for 2 weeks to alleviate HSD-induced renal injury. For the study, 16S rRNA gene sequencing, non-targeted metabonomics, real-time fluorescent quantitative PCR, and Masson’s staining were used to investigate the mechanism of *L. plantarum* ZDY2013 in alleviating renal damage. Results showed that HSD caused intestinal inflammation and changed the intestinal permeability of mice, disrupted the balance of intestinal flora, and increased toxic metabolites (tetrahydrocorticosteron (THB), 3-methyhistidine (3-MH), creatinine, urea, and L-kynurenine), resulting in serious kidney damage. Interestingly, *L. plantarum* ZDY2013 contributed to reconstructing the intestinal flora of mice by increasing the level of *Lactobacillus* and *Bifidobacterium* and decreasing that of *Prevotella* and *Bacteroides*. Moreover, the reconstructed intestinal microbiota significantly changed the concentration of the metabolites of hosts through metabolic pathways, including TCA cycle, ABC transport, purine metabolism, and histidine metabolism. The content of uremic toxins such as L-kynurenine, creatinine, and urea in the serum of mice was found to be decreased by *L. plantarum* ZDY2013, which resulted in renal injury alleviation. Our data suggest that *L. plantarum* ZDY2013 can indeed improve chronic kidney injury by regulating intestinal flora, strengthening the intestinal barrier, limiting inflammatory response, and reducing uremic toxins.

## 1. Introduction

A high-salt diet (HSD) is one in which more than 6 g of salt is consumed per day, including the amount of salt consumed through various sources such as soy sauce, pickles, and monosodium glutamate. Four grams of salt per day is enough for an adult to meet his/her daily physiological needs [1], yet individuals in some Asian countries consume more than 10 g [2]. As a result, metabolic diseases caused by HSD are becoming a major public health problem worldwide. Research confirms the health risks of excessive salt intake, most notably the increased risk of cardiovascular disease associated with HSD [3]. Studies have also found that HSD can affect the body’s immune system and may even increase the risk of autoimmune diseases [4,5]. In addition, HSD can increase the risk of gut related diseases and cancer, as well as affect the kidney system, skeletal system, and metabolic function [6,7]. The incidence of chronic kidney disease (CKD) is on the rise worldwide. High-salt diets are thought to cause kidney damage, but they are still poorly understood. In fact, the presence of CKD is associated with chronic high salt intake. High salt intake leads to tubular damage, which leads to tubular interstitial damage and renal fibrosis [8,9]. In addition, it has been reported that high salt intake has a strong correlation with oxidative stress, which can lead to the oxidation of renal tubule dopamine, resulting in lysosomal and mitochondrial dysfunction and renal inflammation [10]. Meanwhile, it has been pointed out that the fecal flora of mice fed with HSD can independently cause intestinal leakage and kidney damage in mice [11]. HSD can damage the kidney, and the mechanism of the intestinal tract in a HSD-induced kidney injury needs to be further explored and studied.

In previous studies, various drugs and hormones have been aimed at treating or alleviating kidney damage caused by HSD. For example, telmisartan can alleviate HSD-induced kidney injury [12]. A short-term anti-inflammatory treatment with dexamethasone temporarily alleviates fibrosis after chronic kidney injury [13]. These treatments often have side effects; telmisartan can cause abdominal pain, diarrhea, dyspepsia, and other gastrointestinal reactions [14], dexamethasone can cause hyperglycemia, weight change, and metabolic diseases in some patients [15]. Therefore, it is necessary to find an adjuvant therapy with minimal side effects to reduce kidney damage. Lactic acid bacteria are considered as potential probiotic candidates for regulating intestinal flora and improving intestinal barrier integrity [16,17], inhibiting intestinal immunity [18] and intestinal inflammation [19,20]. ZDY2013 (CCTCC M2014170) is a strain of *L. plantarum* screened from homemade fermented soybeans. Previous experiments have found it can improve intestinal microecology, promote intestinal health, and has acid resistance, oxidation resistance, and inflammation inhibition abilities [21,22]. In this study, the effect of *L. plantarum* ZDY2013 on the intestinal tract and on kidney damage was analyzed and compared with that of *Lactobacillus rhamnosus* GG (LGG), one of the best-studied and internationally universal probiotics on the market that has been used safely for many years, alongside integrating multidimensional datasets of the gut microbiome, serum metabolomes, and host characteristics based on previous research data. The purpose of this study was to demonstrate the mechanism of how probiotics alleviate kidney damage caused by a high-salt diet.

## 2. Materials and Methods

### 2.1. Study Design and Sample Collection

Four-week-old specific pathogen-free (SPF) male C57BL/6J mice (Beijing Vital River Laboratory Animal Technology Co. Ltd.) were housed at the animal facility of Jiangxi Academy of Sciences under 12 h light/dark cycles and standard conditions for temperature and humidity. All mice received an irradiated control diet and sterile tap water ad libitum upon arrival. All experimental procedures were according to the National Institutes of Health guidelines and were approved by the local Animal Care and Use Committee of the Nanchang University.

All the purified diets for the control group (NSD group) and the experimental group were purchased from Trophic Animal Feed High-tech Co. Ltd., China. After one week of acclimation in the sterile mouse room, mice in the NSD group (*n* = 10) were given a normal diet and normal drinking water, and mice in the experimental group were given a 4% NaCl high-salt diet with normal drinking water. After 4 weeks, the experimental group was divided into four groups. The HSD group (*n* = 7) was given a 4% NaCl high-salt diet and orally administrated with 200 µL 1 × PBS; the HHZ group (*n* = 8) was given a 4% NaCl high-salt diet and orally administrated with 200 µL 1 × 10^10^ CFU/mL *L. plantarum* ZDY2013; the HLZ group (*n* = 8) was given a 4% NaCl high-salt diet and orally administrated with 200 µL 1 × 10^8^ CFU/mL *L. plantarum* ZDY2013, and the HHL group (*n* = 8) was given a 4% NaCl high-salt diet and orally administrated with 200 µL 1 × 10^10^ CFU/mL *L. rhamnose* GG. The weights of the mice were measured every two days during the experiment. After being gavaged for 2 weeks, the mice were euthanized with ether. The serum, colon, cecum, colon contents, cecum contents, and kidney tissues of the mice were collected in a sterile environment and stored at −80 °C for subsequent experiments.

### 2.2. DNA Extraction from Mouse Feces and 16S rDNA Sequencing Analysis

The genomic DNA of mouse feces was extracted, and the purity and concentration of the DNA were detected by the Shanghai Applied Protein Technology Co. Ltd. (Shanghai, China). Specific primers with a barcode and high-fidelity DNA polymerase were used to amplify the selected V3–V4 variable region by PCR, and the corresponding proportion was mixed according to the sequencing volume requirements of each sample. Target fragments of PCR products were recovered by an AxyPrepDNA gel recovery kit (AXYGEN, Shanghai, China), and the QuantiFluor™-ST Blue Fluorescence Quantification System (Promega, Shanghai, China) was used for the detection and quantification of PCR products. Library construction was performed using the NEB Next^®^ Ultra™ DNA Library Prep Kit. Clean data were obtained by stitching, filtering, and removing chimera from the original data. Then, OTUs (Operational Taxonomic Units) clustering and species classification were analyzed based on available data. Based on the results of OTU cluster analysis, multiple diversity index analysis and sequencing depth detection were conducted for OTU. Based on taxonomic information, the community structure was statistically analyzed at each taxonomic level [23].

### 2.3. H&E and Masson’s Staining

Fresh colon and kidney tissue were collected and soaked in 10% formalin tissue fixative solution. Colonic tissues were processed, including formal in-fixing, paraffin embedding, sectioning, and hematoxylin and eosin (H&E) staining by the Wuhan Barfil Biotechnology Service Co. Ltd. (Wuhan, Hubei, China). Masson’s staining was performed to observe the degree of renal fibrosis. Multiple non-repeating glomerular fields (×200) were selected from each specimen under the same condition. The blue fibrous regions in the glomeruli were regarded as positive targets, and the ratio of the positive area to the total area of the glomeruli was used as the glomerular fibrosis index. The severity of intestinal inflammation was determined by the compactness of the arrangement of intestinal villi, the presence of epithelial cells in the intestinal tract, and the clarity of goblet cells [24].

### 2.4. Protein and Genes Expression

High-quality RNA (OD260/280 and 260/230) were isolated from frozen kidneys and intestinal tissue using the TaKaRa RNA extraction kit (TaKaRaMiniBEST Universal RNA Extraction Kit, Shanghai, China) and then used for cDNA synthesis with a transcriptor cDNA kit (TaKaRa Prime Script^tm^RT reagent kit with gDNA Eraser), according to the manufacturer’s protocol. The PCR reaction system included: 5 µL SYBR Green Mix, 0.4 µL of each primer (10 µM), 1 µL (10 ng/µL) cDNA plus 10 µL ddH2O supplement. The three-step PCR reaction conditions were: preheat at 95 °C for 30 s; then, 95 °C for 5 s, 59 °C for 1 min, 72 °C for 30 s for 40 cycles, annealing at 65 °C for 5 s; then, an ultimate extension at 95 °C for 5 s. The primer sequence information is shown in the Appendix (Appendix A). The mRNA levels of the cytokines (TGF-β1_,_ SOD1, SOD2), the immune-related factor (IFN-γ, CCL4, CCL5, FN-1), and tight junction protein-related genes (tjp1, ocln, clan3) in the intestinal and kidney tissue were measured using real-time PCR.

### 2.5. Determination of the Albumin Level of Fecal Bacteria

The feces of mice were weighed and added to 9 mL 1 × PBS. The samples were homogenized thoroughly with manual homogenizer, centrifuged for about 20 min (2000–3000 rpm), then we collected the supernatant to test. After the sample treatment, the concentration of fecal albumin in the sample was determined by double antibodies with a method according to the steps of an ELISA kit (MEIMIAN, Wuhan, China).

### 2.6. Serum Metabolomic Analysis

A comprehensive analysis of the metabolite changes in mice serum was performed using UHPLC-Q-TOF MS technology at the Shanghai Applied Protein Technology Co. Ltd. The analysis process generally includes sample pretreatment, metabolite extraction, LC-MS full scan detection, data pretreatment, statistical analysis, and differential structure identification. Regarding chromatographic conditions, the samples were separated by Agilent 1290 Infinity LC ultra-high performance liquid chromatography (UHPLC, Shanghai, China) HILIC column. Column temperature was 25 °C. Regarding Q-TOF mass spectrometry conditions, the samples were identified by an AB Triple TOF 6600 mass spectrometer, and the first-order and second-order spectra of the samples were collected. ESI Source conditions were as follows: IonSource Gas1 (Gas1): 40; IonSource Gas2 (Gas2): 80; Curtain gas (CUR): 30; source temperature: 65 °C; IonSapary Voltage Floating (ISVF) ±5000 V (plus or minus). The secondary mass spectrometry was obtained by information-dependent acquisition (IDA), and the high-sensitivity mode was adopted. The declupotency potential (DP): ±60 V (plus or minus two modes); collision energy: 35 ± 15 eV. IDA was set as follows: exclude isotopes within 4 Da; candidate ions to monitor per cycle: 10.

For data processing, the original data were converted into. mzXML format by ProteoWizard, and then the XCMS program was used for peak alignment, retention time correction, and peak area extraction [25]. Accurate mass number matching (<25 PPM) and second-order spectrum matching were carried out through the retrieval of the laboratory and built database. Multidimensional statistical analysis was performed on the data after pareto-scaling pretreatment, including unsupervised principal component analysis (PCA), supervised partial least squares discriminant analysis (PLS-DA), and orthogonal partial least squares discriminant analysis (OPLS-DA). One-dimensional statistical analysis included Student’s *t*-test variance analysis, and R software to draw a volcano map.

### 2.7. Statistical Analysis

Data were expressed as mean ± SD. Statistical analysis was performed with SPSS 19.0 software (SPSS Inc, Chicago, IL, USA) or GraphPad Prism 7.0 (GraphPad Software, Inc, San Diego, CA, USA). A paired-samples *t*-test was performed to analyze the relationships between two groups. Multiple comparisons were evaluated using one-way or two-way ANOVA, followed by Tukey’s multiple-comparison test. *p* < 0.05 was considered statistically significant.

## 3. Results

### 3.1. HSD Broke the Balance of Intestinal Flora, while L. plantarum ZDY2013 Alleviated It

In order to determine the effect of HSD on intestinal microflora composition and the role of *L. plantarum* ZDY2013 in the intestinal tract, we analyzed the microflora structure of mice in the five groups (Figure 1a) by 16S rDNA amplicon sequencing. Based on a 97% sequence similarity, a total of 1898 OTUs were identified in the five groups, among which 716 OTUs were shared by the five groups; NSD, HSD, HHL, HHZ, and HLZ groups contained 64, 113, 94, 97, and 127 unique OTUs, respectively (Figure 1b). The difference in intestinal bacterial community diversity between the HSD group and treated groups were evaluated by α and β diversity analysis. The α diversity of the intestinal microbial population was reflected by the community abundance (ACE) and diversity index (Shannon). Compared with the NSD group, HSD significantly decreased the ACE and Shannon indexes, and the ACE and Shannon indexes in HHL, HHZ, and HLZ groups were significantly higher than those in the HSD group (*p* < 0.05, Figure 1c,d), which indicated that HSD, *L. plantarum* ZDY2013, and LGG significantly altered the intestinal microbial richness and diversity in the mice. Intergroup difference analysis based on unweighted UniFrac β diversity analysis showed that the difference between the HSD group and the NSD group was the largest, while the difference between the HHZ, HHL, HLZ groups and the NSD group was small (Figure 1e). Non-Metric Multi-Dimensional Scaling (NMDS) analysis and PCOA were used to analyze the β-diversity between groups, and the intestinal microbiota distributions between the HSD group, the probiotics treatment groups, and the control groups were clustered separately with each group (Appendix A). These results indicated that HSD, ZDY2013, and LGG changed the richness and diversity of intestinal microflora.

We further analyzed the changes in the cecum microbiota composition at the phylum, genus, and species levels. Lefse and cladogram analysis showed that, compared with the control group (NSD), the relative abundance of Bacteroides in the HSD group increased, while the relative abundance of Firmicutes decreased. In addition, when the ratio of Firmicutes/Bacteroidetes in the HSD group increased significantly, the changes in the Firmicutes/Bacteroidetes ratio were significantly inhibited after the intragastric administration of *L. plantarum* ZDY2013 and LGG, and the structure of the intestinal microflora of mice was more inclined to the NSD group (Figure 1f,g). According to the species annotation results on OTUs, we selected the relative abundance of nine microflora from five groups of mouse stool samples at the bacteria “phylum” level to make a histogram for comparative analysis. Compared with the NSD group, the relative abundance of *Bifidobacterium*, *Lactobacillus*, *Roseburia*, *Allobaculum*, and *Butyricicoccus* in the HSD group was significantly decreased, while the relative abundance of these bacteria was significantly increased in HHL, HHZ, and HLZ groups. In contrast, compared to the NSD group, the relative abundance of *Prevotella* was significantly increased in the HSD group and significantly decreased in the HHL, HHZ, and HLZ groups (Figure 1h).

### 3.2. L. plantarum ZDY2013 Alleviated Body Damage Caused by Long-Term High-Salt Diet

HSD can cause many physiological diseases, and the weight change in mice is important apparent data. The results showed that the weight of mice in the NSD group increased steadily, the weight gain of mice in the HSD group was little, and it was significantly lower compared with those of the NSD group. Compared with the HSD group, the mice orally administrated with different concentrations of *L. plantarum* ZDY2013 and LGG gained significantly more weight, among which the HLZ group gained the most weight, followed by the HHL group and the HHZ group (Figure 2a). H&E staining and Masson’s staining were used to observe the changes in the colon and kidney sections of mice. According to the pathological examination of the colon, compared with the NSD group, the intestinal edge of the HSD group mice was blurred and the inflammatory infiltration was obvious, while after being treated with *L. plantarum* ZDY2013 and LGG, the intestinal edge of the mice in a treated group (HHL, HHZ, and HLZ group) was clear and the inflammation was alleviated (Figure 2b). Likewise, according to the pathological examination of the kidney, compared with the NSD group the mice in the HSD group exhibited severe renal fibrosis, glomerulosclerosis, and renal tubulointerstitial injury, presenting a kidney inflammatory response, which was all alleviated in the treatment groups (Figure 2c,d).

Quantitative real-time PCR (qPCR) experiments were conducted to detect the mRNA expression levels of IFN-γ, CCL4, FN-1, SOD, and TGF-β1 in kidney and colon tissues. The results showed that HSD decreased the mRNA expression levels of TGF-β1 and FN-1, but increased the mRNA expression level of IFN-γ. However, *L. plantarum* ZDY2013 and LGG treatment increased the mRNA expression levels of TGF-β1 and FN-1, decreased that of IFN-γ, and promoted the expression level of TGF-β1, FN-1, and IFN-γ close to that of the NSD group (Figure 2e). In addition, the expression of related cytokines in kidney tissues was detected. The results showed that, compared with the NSD group, the mRNA expression levels of TGF-β1, SOD2, CCL4, and FN-1in the HSD group were all significantly upregulated. After treatment with *L. plantarum* ZDY2013 and LGG, the mRNA expression levels of TGF-β1, SOD2, CCL4, and FN-1 were all significantly downregulated, being closer to the levels of the NSD group (Figure 2f).

To detect the changes in the intestinal permeability of mice, we measured the concentration of fecal albumin and the mRNA expression levels of intestinal tight junction proteins tjp1, ocln, and cldn-3 in mice. Compared with the NSD group, the mRNA expression levels of tjp1, ocln, and cldn-3 were downregulated in the HSD group, while the mRNA expression levels of these proteins were upregulated after treatment with *L. plantarum* ZDY2013 and LGG. Notably, the upregulated expression level in the HLZ group was higher than that of the HHZ group and the HHL group, and closer to the NSD group (Figure 2g). Moreover, the content of fecal albumin in the HSD group was significantly higher than that of the NSD group, while the content of fecal albumin in the treated group was reduced and the HLZ group was closer to the NSD group than the HHL and HHZ groups (Figure 2h). These results indicated that HSD damaged intestinal barrier function while *L. plantarum* ZDY2013 and LGG treatment alleviated intestinal barrier function, among which the HLZ group had the best alleviating effect. 

### 3.3. L. plantarum ZDY2013 Relieved Kidney Injury by Reducing Uremic Toxin

We performed a metabolomic analysis of mouse serum by UHPLC-Q-TOF-MS. The total ion chromatography showed the representative TIC chromatogram of the mouse serum samples, indicating that the variation caused by instrumental errors was small throughout the experiment (Appendix A). In the positive and negative ion modes, the QC samples were closely clustered together, which indicates that the experiment had good repeatability, stability, and reliable experimental data (Appendix A). A principal component analysis (PCA) of serum metabolites showed obvious clustering in each group and a certain distance between groups, indicating that the experiment had good repeatability, significant differences in metabolites between groups, and that the model was stable and reliable (Figure 3a).

One-way and two-way ANOVA was applied to select differential metabolites, a total of 104 significantly different metabolites were observed in the positive and negative ion modes, including THB, taurine, alpha-linolenic acid, vitamin E, 3-MH, citrate, and L-fucose. However, most of the significantly different metabolites were urinary toxins such as creatinine, guanidine, acetic acid, phenols, indoles, canine uric acid, phenol, and advanced glycosylation end products (Appendix A). Volcanic maps can directly show the metabolites that are significantly different between two samples, allowing potential biomarkers to be screened out. The volcanic map of the NSD group and HSD group under a positive ion mode is shown in Figure 3b. The metabolites that were significantly upregulated in the serum of the HSD mice were tetrahydrocorticosteron (THB), 3-methyhistidine (3-MH), creatinine, urea, and L-kynurenine. Significant downregulated metabolites included vitamin E, l-fucose, d-quinovose, 3-indolepropionic acid (IPA), indolelactic acid (ILA), and indoleacetic acid (IAA). 

On the basis of the metabolite cluster heat map of different groups (Figure 3c), HSD increased the serum metabolite levels of THB, urea, creatinine, L-kynurenine, and 3-MH, whereas treatment with *L. plantarum* LGG and ZDY2013 decreased the levels of these metabolites. For all metabolites, we found that most of the metabolites with significant changes were toxic metabolites, and their changes had certain effects on the body. For most toxic metabolites, *L. plantarum* ZDY2013 and LGG alleviated the changes in these metabolites, and the effect of HLZ on toxic metabolites was higher than that of the HHZ group and the HHL group (Appendix A). As for the clinical indicators of kidney injury, serum creatinine, and urea, HSD significantly increased serum creatinine and urea concentrations, while *L. plantarum* ZDY2013 and LGG significantly reduced these concentrations (Figure 3d,e). In addition, HSD decreased serum metabolites such as taurine, indoles, and deoxycholic acid, whereas *L. plantarum* ZDY2013 and LGG increased the levels of these metabolite, being closer to the NSD group. Intestinal microbes are known to metabolize tryptophan to indole metabolites [26]. HSD significantly reduced the serum levels of ILA, IAA, and IPA. However, the contents of ILA, IAA, and IPA in the serum metabolites of the HHL, HHZ, and HLZ groups were increased, and this increase was significant in the HLZ group (Figure 3f–h). We speculated that the indole metabolites changed significantly after the regulation of intestinal flora by a low concentration of *L. plantarum* ZDY2013.

### 3.4. L. plantarum ZDY2013 Alleviates Kidney Injury via Regulating Intestinal Microorganisms to Reduce Urinary Toxin Concentration

In order to investigate the metabolic pathways of significantly different metabolites and the relationship between metabolites and intestinal flora, we performed a joint analysis of intestinal microbiology and metabolomics. The KEGG pathway enrichment of differential metabolites was analyzed by Fisher’s exact test. The results are shown in Figure 4a. In the enrichment analysis of KEGG pathway, a total of 55 KEGG enrichment pathway maps were screened. Among them, the ABC transporter, bile metabolism, purine metabolism, arginine biosynthesis, unsaturated fatty acid biosynthesis, and tryptophan metabolism were the most significant metabolic pathways. The hierarchical clustering heat map showed 975 pairs of different significantly correlated genera and metabolites, of which 543 pairs were more significantly correlated (*p* < 0.01) (Figure 4b). Furthermore, 16S rDNA analyses show that HSD increased the relative abundance of *Desulfosporosinus*, *Rhodanobacter*, *Bacteroides*, and *Prevotella,* and decreased the relative abundance of *Lactobacillus* and *Bifidobacterium* in the intestinal flora. Through a Spearman correlation analysis hierarchical clustering heat map, we found that *Desulfosporosinus*, *Rhodanobacter, Bacteroides,* and *Prevotella* were positively correlated with taurine, hydroxyphenylacetic acid (HPA), 3-MH, and L-kynurenine, and negatively correlated with IAA, ILA, IPA, and dihomo-gamma-linolenic acid. In addition, after treatment of LGG and ZDY2013, the relative abundance of *Desulfosporosinus*, *Rhodanobacter*, *Bacteroides,* and *Prevotella* in the intestinal tract decreased, while that of *Lactobacillus* and *Bifidobacterium* in the intestinal tract increased. Through a Spearman correlation analysis, we found that *Lactobacillus* and *Bifidobacterium* were positively correlated with IAA, ILA, IPA, and dihomo-gamma-linolenic acid, and negatively correlated with taurine, hydroxyphenylacetic acid (HPA), 3-MH and L-kynurenine (Figure 4c, Appendix A). The network map provides a new perspective to study the correlation between significantly different microflora and significantly different metabolites, and the results are similar to the hierarchical clustering heat map (Appendix A). Scatter plots were used to further show the individual associations between related bacteria and metabolite. Figure 4c shows that the relative abundance of Bifidobacterium was positive correlated with ILA and dihomo-gamma-linolenic acid, and the relative abundance of Lactobacillus was positively correlated with IPA and dihomo-gamma-linolenic acid. However, the relative abundance of *Prevotella* was negatively correlated with IPA and dihomo-gamma-linolenic acid, and positively correlated with S-methy-5′-thioadenosine, L-Kynurenine. Therefore, HSD destroyed the intestinal flora of the mice and resulted in metabolic disorders. LGG and ZDY2013 treatment rebuilt the intestinal flora and resulted in the changes in metabolites, which alleviated the metabolic disorders caused by HSD.

## 4. Discussion

With the development of processed foods worldwide, high salt intake has increased significantly, placing people in all systems and organizations under great pressure. As a result, the risk of kidney injury, hypertension, and autoimmune diseases has greatly increased. Batlle reported that mice with a HSD showed less weight gain and more growth delays compared to mice with a low-salt diet [27]. Jane reported that mice fed with a HSD showed changes in their gut microbiome and increased intestinal inflammation [28]. Meanwhile, the low-salt diet improved insulin sensitivity and prevented kidney damage [29]. HSD increased the production of autoantibodies and proinflammatory cytokines and enhanced renal lesions in mice [30,31]. Moreover, HSD also changes the expression of cytokines. Many studies have shown that the increase in FN-1 and TGF-β1 can promote the production of fibroblasts, resulting in the fibrosis of kidney tissue [32,33]. In addition, chronic HSD has been reported to increase intestinal permeability and promote intestinal damage and intestinal bacterial migration across the intestinal barrier to the kidneys, causing kidney damage [11]. The intestinal barrier can prevent pathogenic bacteria, toxins, and other harmful substances from entering the intestine, and damage to the intestinal barrier is thought to be an important factor in bacterial translocation, the presence of bacterial products in systemic circulation, and the state of inflammation [34]. In the present study, both the expression of tight junction protein and the change in fecal albumin concentration indicated that the intestinal permeability of mice in the HSD group was increased. The intestinal permeability of mice in the HHL, HHZ, and HLZ groups was lower than that of the HSD group. H&E staining, Masson’s staining, and cytokines such as TGF-β showed significant fibrosis and inflammatory infiltration in the kidneys of mice in the HSD group. However, fibrosis and inflammation were alleviated in the HHL, HHZ, and HLZ groups. The results suggested that HSD promoted intestinal inflammation by destroying the intestinal barrier, further causing injury, while *L. plantarum* ZDY2013 and LGG may alleviate the damage caused by HSD.

HSD is widely believed to cause an intestinal microecological imbalance and is one of the important causes of the destruction of host health. Studies showed that after chronic HSD feeding, the intestinal flora of mice was significantly changed, the abundance of *Alistipes* and *Ruminococcaceae* UGg-009 in the gut was increased, and that of intestinal *Lactobacillus* was decreased dramatically [35]. Li reported that feeding mice pickles with salt disrupted the metabolism of short-chain fatty acids and the intestinal flora [36]. Similar results were found in our study. A long-term high-salt diet changed the intestinal flora, reducing the relative abundance of *Bifidobacterium*, *Lactobacillus*, *Roseburia*, *Allobaculum*, and *Butyricicoccus*. There is evidence that probiotics are beneficial for gastrointestinal diseases. For example, the probiotic strain *Lactobacillus* casei Zhang (LCZ) isolated from koumiss can alleviate allergic intestinal epithelial injury through NF-kB pathway [37]. The oral administration of fermented milk with Bacillus subtilis (JNFE0126) significantly reduced the disease activity index (DAI) of IBD and balanced the intestinal flora by increasing the abundance of *Bacillus*, *Alistipes,* and *Lactobacillus*, and by decreasing the abundance of *Escherichia* and *Bacteroides* [38]. In addition, the pretreatment of *Bifidobacterium bifidum* CGMCC 15068 can modulate sodium dextran sulfate (DSS)-induced colitis-associated colon cancer (CRC) in mice by increasing the relative abundance of *Akkermansia*, *Desulfovibrionaceae*, *Romboutsia*, *Turicibacter*, *Verrucomicrobiaceae*, *Ruminococcaceae* UCG 013, *Lachnospiraceae* UCG 004, and *Lactobacillus* [39]. Similarly, *L.rhamnose* L34 attenuated the severity of the disease in mice with colitis by regulating intestinal flora to reduce intestinal fungi [40]. In the results of this study, the relative abundances of *Bifidobacterium*, *Lactobacillus*, *Roseburia*, *Allobaculum*, and *Butyricicoccus* were significantly increased after *L. plantarum* ZDY2013 and LGG were administrated. 

Studies indicate that probiotics can rebuild the intestinal microbial community, increase the colonization of probiotics such as *Lactobacillus* in the intestinal tract, and protect and improve the integrity of the intestinal barrier [41,42]. Wan found that *L. plantarum* 69-2 and GOS could activate the hepatic AMPK/SIRT1 signaling pathway by regulating the gut microbiota and metabolites through the liver–gut axis to restore hepatic antioxidant activity in order to alleviate aging [43]. Probiotics regulate the growth of bacteria in the intestinal tract and combine with intestinal toxic molecules to reduce the level of bacterial urea retention in molecules [44]. In addition, probiotics can maintain the weight of mice with chronic malnutrition and increase the production of beneficial metabolites to keep young mice growing [45]. In this research, we found that two weeks after *L. plantarum* ZDY2013 was orally administered, the colonization of *Lactobacillus* and *Bifidobacterium* was selectively promoted in the intestinal tract and the colonization of *Bacteroides* and *Prevotella* was reduced. These effects may be responsible for the decrease in urine toxin production.

The uremia toxin refers to the significant increase in the concentration of body fluid in renal failure patients, and is closely related to the metabolic disorders of uremia or clinical manifestations of some substances. At the same time, metabolic pathways play a key role in linking changes in the body’s metabolites to host health. Studies have shown that the function of ABC transport plays a key role in intestinal microbial products (urotoxins) and nephrotoxicity [46]. In addition, during CKD the renal expression of many ABC transporters was altered at the transcriptional level [47]. In bile secretion, the increased renal excretion of sulfuric acid bile acids is associated with renal tubular reabsorption [48]. Bacteria can use citrate as a sole carbon source. Citrate regulates the synthesis of long-chain fatty acids by regulating acetyl-CoA through the tricarboxylic acid cycle. In the KEGG signaling pathways of purine metabolism and histidine metabolism, we found that deoxyinosine, adenine, IMP, and inosine can affect the biosynthesis of hypoxanthine, thus affecting the biosynthesis of xanthine. Moreover, arginine and proline metabolism can regulate the synthesis of creatinine by regulating the level of guanidine acetic acid. In the current study, the treatment groups (HHL, HHZ, HLZ) mitigated the decrease in these metabolites. Through a Spearman correlation analysis hierarchical clustering heat map, we found that *Desulfosporosinus*, *Rhodanobacter*, *Bacteroides,* and *Prevotella* were positively correlated with taurine, hydroxyphenylacetic acid (HPA), 3-MH, and L-kynurenine, and were negatively correlated with IAA, ILA, IPA, and dihomo-gamma-linolenic acid. *Lactobacillus* and *Bifidobacterium* were positively correlated with IAA, ILA, IPA, and dihomo-gamma-linolenic acid, and were negatively correlated with taurine, hydroxyphenylacetic acid (HPA), 3-MH, and L-kynurenine. We hypothesized that *L. plantarum* ZDY2013 could change the relevant abundance of intestinal flora, thereby changing the metabolite (uremia toxin) concentrations in the intestine and the blood through metabolic pathways such as ABC transport, purine metabolism, histidine metabolism, and finally, by reducing renal injury.

The metabolisms of intestinal flora are strongly correlated with host diseases; many studies inferred the gut microbiota metabolism as a potential predictor of health status and a target for therapeutic intervention. Recent studies have revealed a link between intestinal microorganisms and circulatory metabolites in chronic kidney disease, in which the reduction in *Prevotella* is associated with urea excretion [49]. By modifying the metabolic environment, *Bacteroidetes* could increase metabolites and aggravate intestinal infection [50]. Besides, tryptamine produced by Bacteroides colonized in GF mice accelerates gastrointestinal transport [51]. In gastrointestinal diseases, Bacteroides fragilis produces an enterotoxin, which is an important source of chronic inflammation and has been considered as a risk factor for colorectal cancer [52]. Ore identified new relationships between intestinal flora composition and circulatory metabolites and found that trimethylamine N-oxide is a flora-dependent metabolite that is strongly associated with body health [53]. Devilin described that indophenol sulfate, a uremic solute, is associated with increased mortality in patients with chronic kidney disease [54]. Besides, probiotic metabolites in the gut can regulate the progression of host diseases, such as peptostreptococcus, and can produce tryptophan metabolite indoleacrylic acid, thus promoting an intestinal epithelial barrier function and reducing the inflammatory response [55]. McCarthy and Tillman showed that probiotics such as *Lactobacillus* and *Bifidobacterium* could restore damaged intestinal flora, reduce toxic metabolites in plasma, and reduce intestinal inflammation in mice [56]. There is evidence that probiotics modulate the composition and metabolic function of the gut microbiota, contributing to the remodeling of metabolic profiles and influencing the development of obesity [57]. These studies verified that bacterial metabolites can control the direction of host metabolism and the occurrence of disease, and probiotic intake can promote healthy bacteria to compete with the native gut microbe for growth. In our study, HSD destroyed the intestinal flora and increased the toxic metabolites (taurine, hetrahydrocorticosteron (THB), 3-methyhistidine (3-MH), creatinine, urea, and L-kynurenine) in serum. After an intestinal microflora reconstruction by *L. plantarum* ZDY2013, these toxic metabolites in serum were significantly reduced, thus alleviating renal injury caused by HSD.

## 5. Conclusions

HSD destroyed the structure of intestinal flora of model mice, and then damaged the integrity of intestinal mucosal barrier and increased intestinal permeability. Abnormal intestinal microflora accelerated the biosynthesis of toxic compounds, leading to uremic toxins in plasma and aggravating kidney disease. After 2 weeks of treatment with *L. plantarum* ZDY2013, the relative abundance of *Lactobacillus* and *Bifidobacterium* increased and the intestinal flora of the model mice was reconstructed, which changed the production of metabolites and reduced the level of uremic toxin (hetrahydrocorticosteron (THB), 3-methyhistidine (3-MH), creatinine, urea, and L-kynurenine) in the blood circulation, and, finally, alleviated renal injury (Figure 5). This study first clarified the mechanism of *Lactobacillus plantarum* in alleviating renal injury by analyzing the correlation between intestinal microecology and metabolites, which provided a theoretical basis and the candidate strains for targeted therapy for the treatment of kidney injury-related diseases. 

## Figures and Tables

**Figure 1 nutrients-13-03920-f001:**
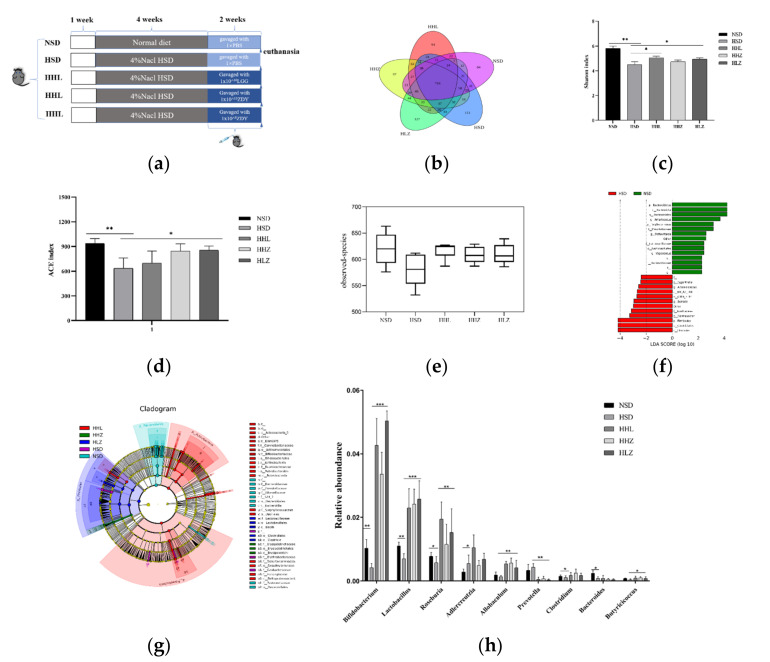
*L. plantarum* ZDY2013 and LGG can alleviate weight loss and intestinal flora disorders caused by high-salt diets. (**a**) A mouse model of chronic renal injury induced by a high-salt diet. (**b**) Petal figure, different colors represent different groups and numbers represent the number of OTUs between the two groups. (**c**) Shannon index. (**d**) ACE index. (**e**) Unweighted UniFrac beta distance analysis of beta-diversity between groups. (**f**) LDA Score, different colors represent different groups and the length of the bar chart represents the size of the LDA value. (**g**) Cladogram, different colors represent different groups, circles represent classification levels from phylum to genus (or species), and small circle diameters are proportional to relative abundance. The red node represents the microorganism group that plays an important role in the red group, the green node represents the microorganism group that plays an important role in the green group, and the yellow node represents the microorganism group that does not play an important role in both groups. (**h**) Histogram of relative abundance of some major bacterial genera. NSD: normal diet group; HSD: 4% high-NaCl diet group; HHL: after a long-term high-salt diet, mice were given 200 µL 1 × 10^10^ CFU/mL *L. rhamnose* by gavage every day; HHZ: after a long-term high-salt diet, mice were gavage with 200 µL 1 × 10^10^ CFU/mL *L. plantarum* ZDY2013 every day; HLZ: after a long-term high-salt diet, mice were gavage with 200 µL 1 × 10^8^ CFU/mL *L. plantarum* ZDY2013 every day. All data are presented as mean ± SD. * *p* < 0.05; ** *p* < 0.01; *** *p* < 0.001.

**Figure 2 nutrients-13-03920-f002:**
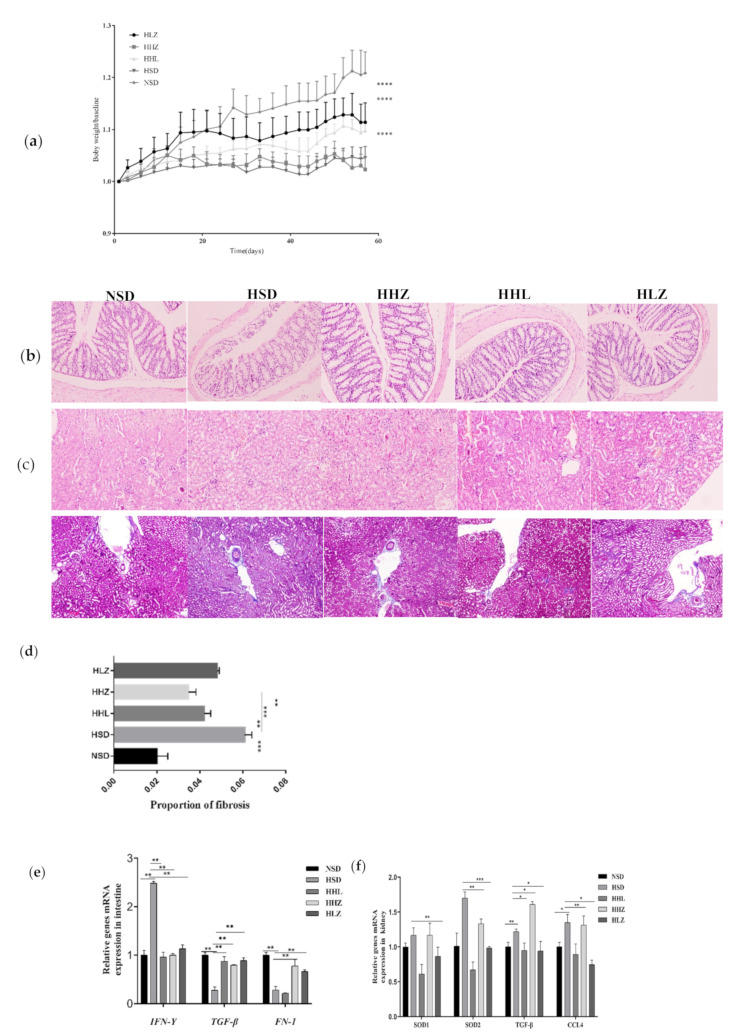
*Lactiplantibacillus plantarum* ZDY2013 can alleviate body damage caused by long-term high-salt diet. (**a**) Changes in body weight over time in each group. (**b**), HE pathological section of mouse colon tissue. (**c**) HE pathological section of mouse kidney tissue. (**d**) The histograms of Masson’s trichromatic staining and the histograms of fibrosis score in mouse kidney tissue. (**e**) Expression of related cytokines mRNA in the intestinal tract of mice. (**f**) mRNA expression of cytokines in kidney tract of mice in different groups. (**g**) Expression of tight junction protein mRNA in the intestinal tract of mice. (**h**) Fecal albumin concentration in mice. All data are presented as mean ± SD. * *p* < 0.05; ** *p* < 0.01; *** *p* < 0.001; **** *p* < 0.0001; paired two-tailed *t*-test.

**Figure 3 nutrients-13-03920-f003:**
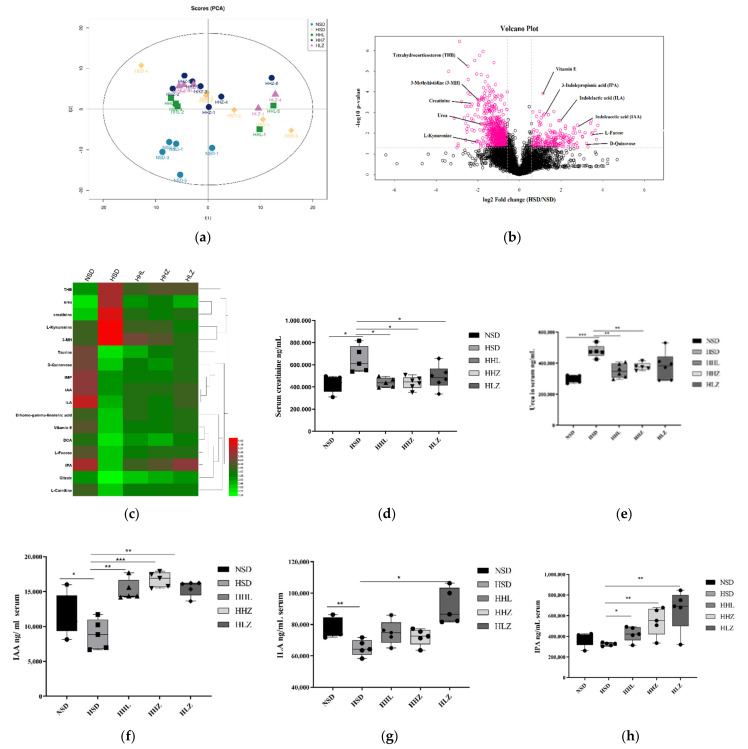
*Lactobacillus plantarum* ZDY2013 can reduce the concentration of urinutoxin increased by HSD. (**a**) Positive and negative ion mode samples of PCA score. (**b**) Volcanic positive ion mode (Volcano Plot), the red dots are differential metabolites screened by univariate statistical analysis. (**c**) Clustering heat map of metabolites with partial significant differences in different groups. (**d**) Serum creatinine concentration in different groups. (**e**) Serum urea concentration in different groups. (**f**–**h**) Serum IAA, ILA, and IPA concentration in different groups. THB: tetrahydrocorticosterone; PAA: phenylacetic acid; IAA: indoleacetic acid; ILA: indolelactic acid; IPA: indolepropionic acid; DCA: deoxycholic acid; IMP: inosine 5′-monophosphate; HPAA: hydroxyphenylacetic acid. All data are presented as mean ± SD. * *p* < 0.05; ** *p* < 0.01; *** *p* < 0.001. Square: HSD group; circle: NSD group; triangle: HHL group; upside down triangle: HHZ group; rhombus: HLZ group.

**Figure 4 nutrients-13-03920-f004:**
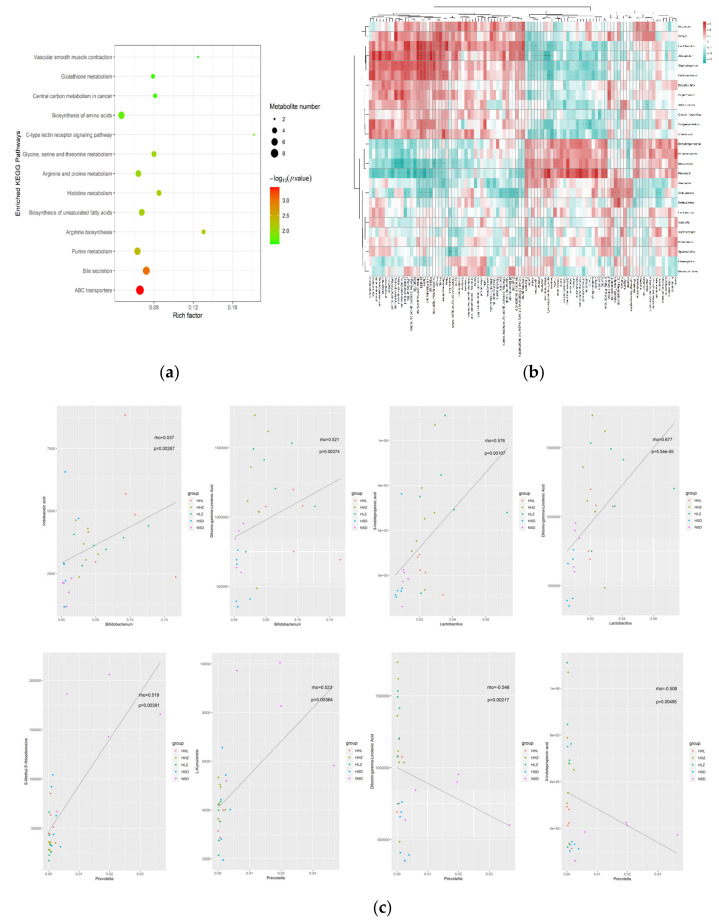
*Lactobacillus plantarum* ZDY2013 alleviates kidney injury by regulating intestinal microorganisms to reduce urinary toxin concentration. (**a**) Results of enrichment analysis of positive ion mode KEGG pathway and KEGG pathway diagram. (**b**) Spearman correlation hierarchical cluster analysis was performed for the significantly different flora and metabolites. Red means positive correlation, blue means negative correlation; the darker the color, the stronger the correlation. (**c**) Spearman correlation network analysis was performed for the significantly different flora and metabolites. Red represents positive correlation, blue represents negative correlation, the thickness of the line is in direct proportion to the absolute value of correlation coefficient, and the node size is in positive correlation with its degree. All data are presented as mean ± SD.

**Figure 5 nutrients-13-03920-f005:**
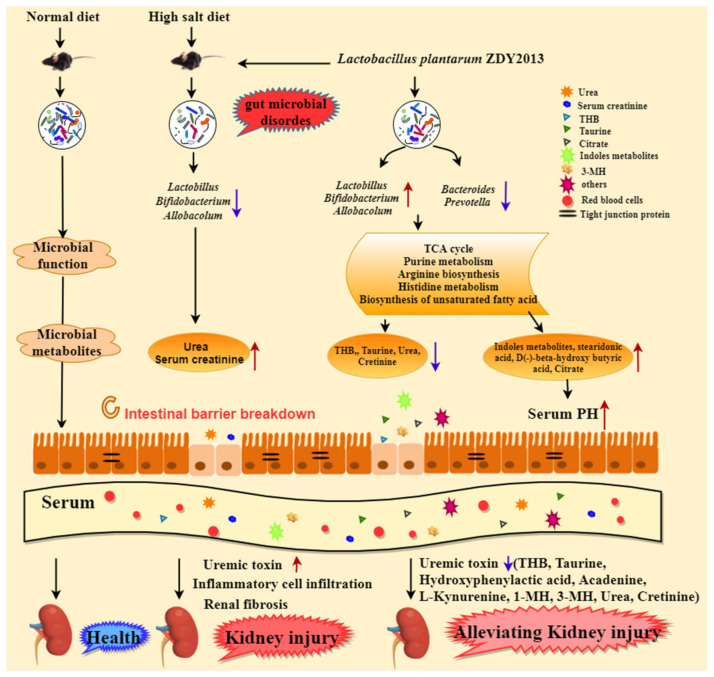
Mechanism figure of *Lactobacillus plantarum* ZDY2013 in alleviating renal injury induced by high-salt diet. The blue downward arrow indicates a decrease in relative abundance or metabolite concentration, while the red upward arrow indicates an increase in relative abundance or metabolite concentration. Different shaped species represent different metabolites, proteins, or cells, etc.

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
