# Peer review of "Serum Untargeted Metabolism Reveals the Mechanism of L. plantarum ZDY2013 in Alleviating Kidney Injury Induced by High-Salt Diet"

_nutrients, 2021, doi:10.3390/nu13113920_

Round 1
Reviewer 1 Report
Manuscript Number: NUTRIENTS-1405663
Title: Serum untargeted metabolism revealed the mechanism of L. plantarum ZDY2013 in alleviating kidney injury induced by high salt diet
In this study, the authors have attempted to evaluate the risks for hypertension and kidney injury associated with high salt diets (HSD) by using advance techniques like real-time fluorescent quantitative PCR, and Masson’s staining. The mice serum results have demonstrated a decrease in uremic toxins by L. plantarum ZDY2013, which resulted in renal injury alleviation., hence suggesting that L. plantarum ZDY2013 can improve chronic kidney injury. The detailed comments described for this article can be considered as “minor revisions” and given in different parts as per sequence of the paper to be considered for publication.
Abstract: It would be nice to revise the abstract by adding principal results of key parameters e.g., concentration of metabolites and blood serum profiles. It will be more informative and interesting for the readers.
Key words: Seem appropriate and represent the content of this paper.
Introduction: Although introduction is concise and appropriate and various literature studies have been mentioned to support their topic, however few questions need to be clarified for further improving the manuscript. While reviewing scientific studies, the authors have mentioned that various drugs and hormones have been aimed at treating or alleviating kidney damage caused by HSD have some side effects (lines 53-56), which ones? It is suggested to highlight those side effects which will equally justify the objectives of current study. Furthermore, it would be equally good to explain (1) how the current study tackles this problem in more effective way as compared to all these previous studies? (2) Why was it necessary to study the effect of L. plantarum while it was previously evaluated to improve intestinal microecology (lines 61-64) through experiments by other scientists?
Materials and methods: The experimental design is appropriate and most of the methods are described in a better way, however, it would be nice to either cite the scientific reference or explain the choice of groups and quantity of oral administration for various groups. Furthermore, none of the methods has been referenced except 2.3, so it will be more appropriate if the methods employed in current study are correctly referred to the scientific protocols which were considered as basis.
Few minor changes are also suggested e.g., please rectify the sentence “then we will send the sample to” by “the samples were sent to” in line 110. The symbol used for liter while writing “μl” in lines 123-124 needs to be rectified by writing “μL” as mentioned in other parts of the methodology e.g., lines 86-89.
Results and discussion: The results have been presented and discussed in better way. The statistical methods and approach to discuss these results along through comparison with literature are appropriate.
Conclusion: Conclusion needs to be improved by adding pertinent results as suggested for abstract.
The detailed comments have been attached as separate file.

Author Response
Abstract: It would be nice to revise the abstract by adding principal results of key parameters e.g., concentration of metabolites and blood serum profiles. It will be more informative and interesting for the readers.
Response 1:
- Thanks for your good suggestion. In the abstract section, we have added the details of toxic metabolites (line 19,20) to highlight which toxic metabolites increased by a long-term high-salt diet caused kidney damage.
Introduction: Although introduction is concise and appropriate and various literature studies have been mentioned to support their topic, however few questions need to be clarified for further improving the manuscript. While reviewing scientific studies, the authors have mentioned that various drugs and hormones have been aimed at treating or alleviating kidney damage caused by HSD have some side effects (lines 53-56), which ones? It is suggested to highlight those side effects which will equally justify the objectives of current study. Furthermore, it would be equally good to explain (1) how the current study tackles this problem in more effective way as compared to all these previous studies? (2) Why was it necessary to study the effect of L. plantarum while it was previously evaluated to improve intestinal microecology (lines 61-64) through experiments by other scientists?
Response 2:
- Thanks for your kind suggestion. We mentioned in the article that telmisartan and dexamethasone are designed to treat or reduce kidney damage caused by HSD, and these treatments often have side effects. Telmisartan can cause gastrointestinal reactions such as abdominal pain, diarrhea and indigestion[1], while dexamethasone can cause high blood sugar, weight changes and metabolic diseases in some patients[2].Their side effects have been added in the article (lines 57-59).
- Thanks for your kind suggestion. In the past, few researches focused on the effect of probiotics alleviating kidney damage. As a food-grade probiotics, Lactobacillus plantarum is recognized as a safe and reliable strain. It has been used as an adjuvant in the treatment of many clinical studies[3,4], but it is rarely used to regulate kidney injury. Therefore, we designed and conducted this research.
- Thanks for your kind suggestion. Although the improvement of intestinal microecology by Lactobacillus plantarum has been studied, few studies have been conducted on the mechanism of Lactobacillus plantarum to alleviate kidney injury through the regulation of intestinal flora and serum metabolites. In addition, Lactobacillus plantarum ZDY2013 is a potential probiotic strain screened by our research group in the early stage, which can regulate intestinal flora, improve gastrointestinal health and of the host, and maintain a high survival rate under acid pressure[5,6]. Therefore, the experimental study took Lactobacillus plantarum ZDY2013 as the research object to explore its alleviating effect on body injury induced by high salt diet.
Materials and methods: The experimental design is appropriate and most of the methods are described in a better way, however, it would be nice to either cite the scientific reference or explain the choice of groups and quantity of oral administration for various groups. Furthermore, none of the methods has been referenced except 2.3, so it will be more appropriate if the methods employed in current study are correctly referred to the scientific protocols which were considered as basis. Few minor changes are also suggested e.g., please rectify the sentence “then we will send the sample to” by “the samples were sent to” in line 110. The symbol used for liter while writing “μl” in lines 123-124 needs to be rectified by writing “μL” as mentioned in other parts of the methodology e.g., lines 86-89.
Response 3:
- Thanks for your kind suggestion. We have cited the scientific references in methods 2.2 and 2.6 (lines 111, 160). In addition, the dosing for different groups was added (lines 87-92).
- Thanks for your kind suggestion. the sentence was modified, the writing “μl” has be instead of “μL” (lines 128-131). Besides, the H&E colonic staining method in 2.3 (lines 114-116).and the reaction system and procedure in real-time quantitative PCR in 2.4 were modified (lines 128-131).
Results and discussion: The results have been presented and discussed in better way. The statistical methods and approach to discuss these results along through comparison with literature are appropriate.
Conclusion: Conclusion needs to be improved by adding pertinent results as suggested for abstract.
Response 4:
- Thanks for your kind suggestion. In the conclusion part, we added some description of the result at line 532-535.

Reviewer 2 Report
Dear authors, it is an interesting paper. Due to the fine description of the experiments and the results, it could be considered that the discussion should be more focused on the results and their implications.
Author Response
Due to the fine description of the experiments and the results, it could be considered that the discussion should be more focused on the results and their implications.
Response :
- Thanks for your kind suggestion. Indeed, for the discussion section, we should focus more on the results and their implications. Therefore, the discussion on the correlation analysis between intestinal microflora and metabolites is increased, and the strong correlation between intestinal microflora and serum metabolites is emphasized. It is concluded that LGG and ZDY2013 may regulate the generation of metabolites through intestinal microflora reconstruction after treatment, thus alleviating metabolic disorders caused by HSD (lines 484-489). And at the end of the discussion, we added some implications of the result at line 517-521.
